# Mind Your Solver! On Adversarial Attack and Defense for Combinatorial Optimization

## Abstract

Combinatorial optimization (CO) is a long-standing challenging task not only in its inherent complexity (e.g. NP-hard) but also the possible sensitivity to input conditions. In this paper, we take an initiative on developing the mechanisms for adversarial attack and defense towards combinatorial optimization solvers, whereby the solver is treated as a black-box function and the original problem's underlying graph structure (which is often available and associated with the problem instance, e.g. DAG, TSP) is attacked under a given budget. Experimental results on three real-world combinatorial optimization problems reveal the vulnerability of existing solvers to adversarial attack, including the commercial solvers like Gurobi. In particular, we present a simple yet effective defense strategy to modify the graph structure to increase the robustness of solvers, which shows its universal effectiveness across tasks and solvers.

## 1 Introduction

The combinatorial optimization (CO) problems are widely studied due to their importance in practice (e.g. job scheduling, routing, matching, etc). In the last century, a variety of heuristic methods (Van Laarhoven & Aarts, 1987; Whitley, 1994) are proposed to tackle these standing and often NP-hard problems. Driven by the recent development of deep learning and reinforcement learning, many learning-based methods (Khalil et al., 2017; Mao et al., 2019; Kwon et al., 2021) are also developed in this area, which show promising potential often for their cost-efficiency.

Despite the success of solvers in various combinatorial optimization tasks, few attention has been paid to the vulnerability and robustness of combinatorial solvers, regardless of whether they are learning based or not. A line of relevant works aims at handling combinatorial optimization under uncertainty (Buchheim & Kurtz, 2018). However, to our best knowledge, ensuring the robustness of combinatorial solvers with slightly modified problem instances remains relatively unexplored. It is worth noting that many CO problems can be essentially formulated as a graph problem (Khalil et al., 2017; Bengio et al., 2020), hence it is attractive and natural to modify the problem instance by modifying the graph structure, to generate more test cases for solvers. In fact, vulnerability can often be an inherent challenge for CO solvers since the problem is often strong nonlinear and NP-hard. From this perspective, we consider attack and defense CO solvers in the following aspects.

From the attack side, developing attack models can be useful for thoroughly evaluating a solver's robustness. The solvers may be more fragile than the general impression: for traditional learning-free solvers, in some cases, their heuristics and hyperparameters may not be universal and stable enough such that a small change on problem condition or graph structure may deteriorate the performance notably. This also holds for recent machine learning based solvers as the model may be overfit and the objective landscape can be complex due to the inherent difficulty of discrete CO problems.

As a result, it is imperative to develop defense mechanisms and techniques to improve the robustness of CO solvers, either for learning-based models or traditional ones, especially if the approach can be in black-box mode without knowing the details of the solvers. In particular, it is even desirable to develop out-of-box defense mechanism. Our hope is that this may be realized when the problem instance change[1] involves only graph structure variation – which is often the case.

---

[1] Readers may argue that there are little deliberate attacks to CO solvers, while one can regard such attacks as the problem instance variation which can often happen in real-world e.g. when the network takes a small daily change in Directed Acyclic Graph (DAG) and Fraud Coverage problems as will be studied in our experiments.

Table 1: Comparing our framework (ROCO) with FGSM (Goodfellow et al., 2015) and RL-S2V (Dai et al., 2018). $\epsilon$-perturb. means the change of one pixel should be bounded in $\epsilon$. B-hop neighbourhood means the new attack edges can only connect two nodes with distance less than $B$.

| Method | Data | Task | Attack target | Attack cost | Attack principle | Defense tech. |
|--------|------|------|--------------|-------------|-----------------|---------------|
| FGSM | image | classification | pixels | $\epsilon$-Perturb. | invisible change | adversarial Training |
| RL-S2V | graph | classification | edges (connectivity) | edge # | B-hop neighbour | random drop |
| ROCO | CO instance | CO solution | edges (constraints) | edge # | no worse optimum | symmetric RL |

To this end, we present **Ro**bust **C**ombnaotorial **O**ptimization (ROCO), a framework for testing and improving the robustness of a given combinatorial optimization solver. Table 1 compares our framework to classical works in images and graphs. Our attacker limits the number of attacked edges in the graph and guarantees that the optimal solution must not become worse. Our defender ensures that the new solution is also feasible for the pre-defended problem. The overview of ROCO framework is summarized in Fig. 1. **In summary, this paper makes the following contributions**:

**1)** Given the fact that combinatorial problems can often be represented by underlying graphs, we propose to perform adversarial attacks toward CO solvers to deteriorate their solution quality. To our best knowledge, this is the pioneering work that formally studies adversarial attacks on combinatorial solvers, though their vulnerability has been occasionally recognized by the community.

**2)** We propose ROCO, an adversarial framework that consists of both attack and defense models on top of CO solvers. We design our attack models with both learning-based and traditional simulated annealing methods by slightly modifying the graph structures (e.g. add, delete or modify edges). To increase the robustness of the combinatorial solvers, we further propose defense mechanism against attacks. Our attack and defense models are applicable to solvers regardless of learning-based or not.

**3)** We implement and apply our adversarial attack and defense models to three common combinatorial optimization tasks: Directed Acyclic Graph Scheduling, Asymmetric Traveling Salesman Problem and Fraud Coverage. The experimental results on black-box attack/defense show the effectiveness and generality of our approach. The source code will be made public available.

## 2 RELATED WORK

**Combinatorial optimization.** As a widely studied problem, there exist many traditional algorithms for CO, including but not limited to greedy algorithms, heuristic algorithms like simulated annealing (SA) (Van Laarhoven & Aarts, 1987) or Lin–Kernighan–Helsgaun (LKH3) (Helsgaun, 2017), as well as commercial solvers like Gurobi (Gurobi Optimization, 2020). Besides, driven by the recent development of deep learning and reinforce learning, many learning-based methods have also been proposed to tackle these problems. A mainstream approach using deep learning is to predict the solution end-to-end, such as the supervised model Pointer Networks (Vinyals et al., 2015), reinforcement learning models S2V-DQN (Khalil et al., 2017) and MatNet (Kwon et al., 2021). Though these methods did perform well on different types of COPs, they are not that robust and universal, as discussed in (Bengio et al., 2020), the solvers may get stuck around poor solutions in many cases. Different from works (Moon et al., 2019; Zang et al., 2020) which apply CO for attack against neural networks, we take an initiative on the adversarial attack and defense on CO.

**Adversarial attack and defense.** Since the seminal study (Szegedy et al., 2014) showed that small input perturbations can change model predictions, many adversarial attack methods have been devised to construct such attacks. In general, adversarial attacks can be roughly divided into two categories: white-box attacks with access to the model gradients, e.g. (Goodfellow et al., 2015; Madry et al., 2018; Carlini & Wagner, 2017), and black-box attacks, with only access to the model predictions, e.g. (Ilyas et al., 2018; Narodytska & Kasiviswanathan, 2016). Besides image and text adversarial attacks (Jia & Liang, 2017), given the importance of graph-related applications and the successful applications of graph neural networks (GNN) (Scarselli et al., 2008), more attentions are recently paid to the robustness of GNNs. In the mean time, many defense strategies like adversarial training (Ganin et al., 2016; Tramèr et al., 2020) have also been proposed to counter this series of attack methods. Since CO problems can usually be encoded by a graph structure and inspired by (Dai et al., 2018), which develops an RL based attack policy towards GNNs, we propose a novel and flexible attack and defense framework for CO solvers using both heuristic and RL methods.

Note that the recent adversarial graph matching (GM) network show how to fulfill attack or defense via perturbing or regularizing geometry property on the GM solver. (Zhang et al., 2020) degrades the

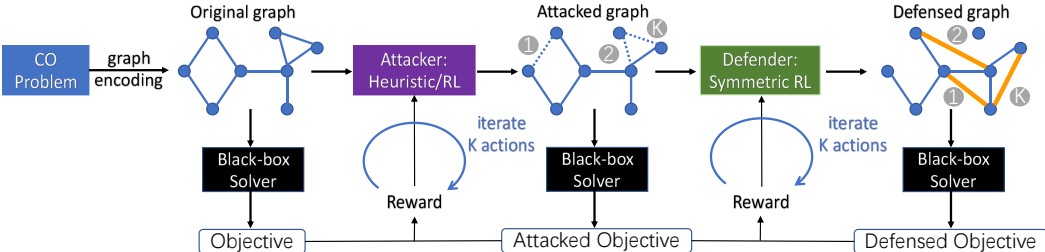

Figure 1: Overview of our attack and defense framework ROCO for CO solvers. ROCO targets on the CO problems which can be encoded by graph (often holds in practice). Here delete/add the edges in the encoded graph represents delete/add constraints in CO. Symmetric RL denotes that the defender and attacker share the same structure with symmetric reward and action space.

quality of GM by perturbing nodes to more dense regions while (Ren et al., 2021) improves robustness by separating nodes to be distributed more broadly. However, the techniques are deliberately tailored to the specific problem and can hardly generalize to the general CO problems. Meanwhile they work in a white box mode while we aim to develop more flexible black box models.

## 3 COMBINATORIAL OPTIMIZATION WITH ATTACK AND DEFENSE

### 3.1 PROBLEM FORMULATION

In general, a traditional CO problem $Q$ defined on graph $\mathcal{G} = (V, E)$ can be formulated as:

$$Q : \min_{\mathbf{x}} f(\mathbf{x}|\mathcal{G}) \qquad s.t. \quad h_i(\mathbf{x}, \mathcal{G}) \leq 0, \ i = 1, \ldots, I \qquad (1)$$

where $\mathbf{x}$ denotes the decision variable, $f(\mathbf{x}|\mathcal{G})$ represents the target function w.r.t. the specific CO problem and $h_i(\mathbf{x}, \mathcal{G})$ denotes the set of constraints (usually encoded in graphs). However, due to the NP-hard nature (which is often the case in CO), it can be infeasible to find the optimal solution within polynomial time. Therefore, we denote a different solver $\mathcal{S}$ (which gives the feasible solution $f(\mathcal{S}(Q)|\mathcal{G})$) to approach the global optimum $f^*(Q)$.

It is worth noting that the optimum $f^*(Q)$ of Eq. 1 will become no worse if we loosen part of the constraints $h_i$ since the previous decision variable $\mathbf{x}$ is still feasible under the new setting. Intuitively, we may expect the solver to give a better (at least the same) solution on the new problem $Q'$. However, we will show in this paper that many solvers are vulnerable to such perturbations and their solutions can become worse under our attacks, despite the loose bound $f^*(Q') \leq f^*(Q)$.

Given a solver $\mathcal{S}$ and an original problem $Q$ represented by a graph $\mathcal{G}$, the adversarial attacker $g$ is asked to modify the graph $\mathcal{G}$ into $\mathcal{G}'$ to attack the solver $\mathcal{S}$, such that:

$$\begin{aligned} \max_{\mathcal{G}'} \quad & f(\mathcal{S}(Q')|\mathcal{G}') - f(\mathcal{S}(Q)|\mathcal{G}) \\ s.t. \quad & \mathcal{G}' = g(\mathcal{S}, \mathcal{G}), \text{hence } Q \rightarrow Q', \qquad f^*(Q') \leq f^*(Q), \quad \mathcal{T}(\mathcal{G}, \mathcal{G}') = 1 \end{aligned} \qquad (2)$$

Here $\mathcal{T}(\cdot, \cdot) \rightarrow \{0, 1\}$ is an equivalency indicator (Dai et al., 2018) that tells whether two graphs $\mathcal{G}$ and $\mathcal{G}'$ satisfy a specified constraint. In short, the above equation tells that the attacker is aiming at making small modifications to the original graph, loosening the constraints while making the solver solution as bad as possible.

In this paper, concretely our attacker $g$ is allowed to modify edges (e.g. adding or removing edges) from $\mathcal{G}$ to construct the new graph. Accordingly, we define the equivalency indicator as:

$$\mathcal{T}(\mathcal{G}, \mathcal{G}') = \mathbb{I}(|(E - E') \cup (E' - E)| \leq K) \qquad (3)$$

which ensures that the attacker can modify no more than $K$ edges of the original graph.

On the other hand, it is imperative to develop defense mechanism for against the above attacks. Notice that the attack methods we mentioned before have some degree of symmetry (adding/deleting edges), we can simply do reverse operations for defense. For example, if we can relax the constraints by removing edges while worsening the solver's solution, then we can add some edges (constraints) and get a better solution (that is, the symmetry). Besides, the new solution under stronger constraints

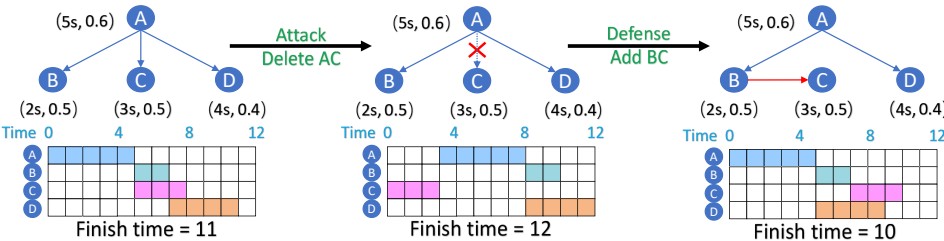

Figure 2: Attack and defense on applying Shortest Job First algorithm for solving DAG. The edges show the dependencies. $(x, y)$ of each node means run time $(x)$ and resource occupancy rate $(y)$.

is surely feasible for the original graph (then we can use it in the original graph to get $f(\mathcal{S}(Q')|\mathcal{G})$). Hence, the new problem can be formulated as:

$$
\begin{aligned}
\min_{\mathcal{G}'} \quad & f(\mathcal{S}(Q')|\mathcal{G}) - f(\mathcal{S}(Q)|\mathcal{G}) \\
s.t. \quad & \mathcal{G}' = d(\mathcal{S}, \mathcal{G}), \text{hence } Q \to Q', \quad H_j(\mathcal{G}', \mathcal{G}) \leq 0, \text{for } j = 1 \dots J, \quad \mathcal{T}(\mathcal{G}, \mathcal{G}') = 1
\end{aligned}
\tag{4}
$$

here the constraints $H_j(\mathcal{G}', \mathcal{G}) \leq 0$ ensure that the feasible space of $\mathcal{G}'$ is a subset of $\mathcal{G}$.

**Concrete Examples.** Fig. 2 shows the attack and defense of the Shortest Job First algorithm on DAG (TSP and FC examples are in Appendix A). We remove an edge but get a worse finish time (objective – the smaller the better). Then we add an edge for defense, which leads to a better solution.

In this paper, we focus on black-box attack and defense, which means we have no idea on the solver. This setting is practical especially considering there are plenty of commercial solvers e.g. Gorubi and CPLEX etc. We leave white box attack and defense for future work.

## 3.2 ATTACK VIA GRAPH MODIFICATION

We devise both reinforce learning (RL) and heuristic based attackers. For RL, the popular Proximal Policy Optimization (PPO) (Schulman et al., 2017) framework is adopted. We also design three traditional heuristic attackers: random sampling, optimum-guided search and simulated annealing.

### 3.2.1 REINFORCE LEARNING BASED ATTACK

Eq. 2 is treated as the learning objective and we resort to reinforcement learning (RL) to optimize $\mathcal{G}'$ in a data-driven manner. In general, we modify the graph structure and compute $f(\mathcal{S}(Q')|\mathcal{G}')$ alternatively, getting rewards that will be fed into the PPO framework and train the agent iteratively.

**MDP Formulation.** Given an instance $(\mathcal{S}, \mathcal{G})$, with a total modification budget, we model the attack via sequential edge modification as a Finite Horizon Markov Decision Process (MDP).

- State. The current graph $\mathcal{G}^k$ (i.e. the graph $\mathcal{G}'$ after $k$ actions) is treated as the state, whose nodes and edges encode both current input and constraints. The original graph $\mathcal{G}^0$ is the starting state.

- Action. As mentioned in Sec. 3.1, the attacker is allowed to add/delete edges in the graph. So a single action at time step $k$ is $a^k \in \mathcal{A}^k \subseteq E^k$. Here our action space $\mathcal{A}^k$ is usually a subset of all the edges $E^k$ because we restrict the action space (i.e. abandon some useless edge candidates) according to the previous solution $\mathcal{S}(Q^k)$ to speed up our algorithm. Furthermore, we decompose the action space $(O(|V|^2) \to O(|V|))$ by transforming the edge selection into two node selections: first selecting the starting node, then the ending node.

- Reward. The new graph $\mathcal{G}^{k+1}$ results in a new CO problem $Q^{k+1}$ whose objective becomes $f(\mathcal{S}(Q^{k+1})|\mathcal{G}^{k+1})$. The reward is the increase of the objective:

$$
r = f(\mathcal{S}(Q^{k+1})|\mathcal{G}^{k+1}) - f(\mathcal{S}(Q^k)|\mathcal{G}^k)
\tag{5}
$$

- Terminal. Once the agent modifies $K$ edges or edge candidates become empty, the process stops.

**PPO Design.** The input and constraints of a CO problem are usually tightly encoded in the graph structure. Thus, our PPO agent (i.e. the actor and the critic) should behave according to the graph features. Specifically, We resort to the Graph Neural Networks (GNN) for graph embedding:

$$
\mathbf{n} = \text{GNN}(\mathcal{G}^k), \; \mathbf{g} = \text{AttPool}(\mathbf{n})
\tag{6}
$$

where the matrix $\mathbf{n}$ (with the size of node number $\times$ embedding dim) is the node embedding, and an attention pooling layer is used to extract a graph level embedding $\mathbf{g}$. The GNN model can differ by the CO problem. After graph feature extraction, we design the corresponding actor and critic net:

- Critic. The critic predicts the value of each state $\mathcal{G}^k$. Since it aims reward maximization, a max pooling layer is adopted over all node features which are concatenated (denoted by $[\cdot||\cdot]$) with the graph embedding $\mathbf{g}$, fed into a network (e.g. ResNet block (He et al., 2016)) for value prediction:

$$\mathcal{V}(\mathcal{G}^k) = \text{ResNet}_1([\text{MaxPool}(\mathbf{n})||\mathbf{g}]) \tag{7}$$

- Actor. As mentioned in Sec. 3.2.1, the edge selection is implemented by selecting the start and end node. The action scores are computed using two independent ResNet blocks, and a Softmax layer is added to regularize the scores into probabilities within $[0, 1]$ as follows:

$$P(a_1) = \text{softmax}(\text{ResNet}_2([\mathbf{n}||\mathbf{g}])), \ P(a_2|a_1) = \text{softmax}(\text{ResNet}_3([\mathbf{n}||\mathbf{n}[a_1]||\mathbf{g}])) \tag{8}$$

where $\mathbf{n}[a_1]$ denotes the embedding for node $a_1$. We add the feature vector of the selected start node for the end node selection. For training, actions are sampled w.r.t. their probabilities. For testing, beam search is adopted to find the optimal solution: actions with top-$B$ probabilities are chosen for each graph in the last time step, and only those actions with top-$B$ rewards will be reserved for the next search step (see Alg. 1 for details).

### 3.2.2 HEURISTIC ALGORITHM ATTACKING

Traditional heuristic algorithms are also studied, with three attack algorithms as follows.

**Random sampling.** In each iteration, an edge is randomly chosen to be modified in the graph and it repeats for $K$ iterations. We run $N$ attack trials and choose the best solution. It can reflect the robustness of solvers with the cost of time complexity $O(NK)$.

**Optimum-guided search (OG-Search).** It focuses on finding the optimum solution during each iteration. We use beam search to maintain the best $B$ current states and randomly sample $M$ different actions from the candidates to generate next states. The number of iterations is set to be no more than $K$. Its time complexity is $O(BMK)$.

**Simulated Annealing (SA).** Simulated annealing (Van Laarhoven & Aarts, 1987) comes from the idea of annealing and cooling used in physics for particle crystallization. In our scenario, a higher temperature indicates a higher probability of accepting a worse solution, allowing to jump out of the local optimum. As the action number increases and the temperature decreases, we will be more conservative and tend to reject the bad solution. The detailed process is shown in Appendix B and we will repeat the algorithm for $N$ times. SA is a fine-tuned algorithm and we can use grid search to find the best parameter to fit the training set. Its time complexity is $O(NMK)$.

Table 2 concludes the attacking methods property and time complexity. Since the former three algorithms are inherently stochastic, we will run them multiple times to calculate the mean and standard deviation for fair comparison.

---

**Algorithm 1: Attack framework by iterative edge manipulation (RL version)**

**Input:** Input graph $\mathcal{G}$; solver $\mathcal{S}$; max number of actions $K$; beam size $B$.

$\mathcal{G}^0_{1..B} \leftarrow \mathcal{G}; \mathcal{G}^* \leftarrow \mathcal{G}$; # set initial state
**for** $k \leftarrow 1..K$ **do**
  **for** $b \leftarrow 1..B$ **do**
    # do beam search for graphs in last step
    Predict $P(a_1), P(a_2|a_1)$ on $\mathcal{G}^{k-1}_b$;
    Select $(a_1, a_2)$ with top-$B$ probabilities;
  **for** each $(b, a_1, a_2)$ pair **do**
    $\mathcal{G}'(b, a_1, a_2) \leftarrow$ modify edge $(a_1, a_2)$ in $\mathcal{G}^{k-1}_b$; # new state by tentative action
    **if** $f(\mathcal{S}|\mathcal{G}'(b, a_1, a_2)) > f(\mathcal{S}|G^*)$ **then**
      $\mathcal{G}^* \leftarrow \mathcal{G}'(b, a_1, a_2)$ # update the optimal attacked graph
  Sort $\mathcal{G}'(\cdot, \cdot, \cdot)$ w.r.t. their solutions by decreasing order; # select top-$B$ graphs for next step
  $\mathcal{G}^k_{1..B} \leftarrow \mathcal{G}'_{1..B}$;

**Output:** Optimal Attacked Graph $\mathcal{G}^*$

---

Table 2: Comparison of attack models. Random means it will produce different results in different trials. Finetune means the algorithm can be tuned by training set.

| Technique | Random | Finetune | Time |
|---|---|---|---|
| Random | ✓ | | $O(NK)$ |
| OG-Search | ✓ | | $O(BMK)$ |
| SA | ✓ | ✓ | $O(NMK)$ |
| RL | | ✓ | $O(BMK)$ |

### 3.3 DEFENSE VIA GRAPH MODIFICATION

We adopt RL as the defender and treat Eq. 4 as the learning objective. The defense MDP formulation is just the same as Sec. 3.2.1 except that we set $r = f(\mathcal{S}(Q^k)|\mathcal{G}) - f(\mathcal{S}(Q^{k+1})|\mathcal{G})$ and use the symmetric action of the attacker. It is worth noting

Table 3: **DAG attack results of Ratio (%)** $\uparrow \pm$ **Std.** Baseline denotes mean finish time (real time should $\times 5000$) on test set. Ratio represents time improvement after attack w.r.t. baselines. The larger the ratio, the better attack performance the adversarial attack method achieve. Random, OG-search, SA are tested for 10 trials to calculate the mean and std.

| Solver | TPC-H job# | Baseline | Attack Method (Ratio $\pm$ Std) | | | |
|---|---|---|---|---|---|---|
| | | | Random | OG-Search | SA | RL |
| Shortest Job First | 50 | 20.9228 | $1.08 \pm 0.12$ | $1.33 \pm 0.18$ | $\mathbf{1.54 \pm 0.07}$ | 1.41 |
| Critical Path | 50 | 17.3900 | $8.13 \pm 0.44$ | $9.03 \pm 0.25$ | $\mathbf{9.58 \pm 0.15}$ | 9.26 |
| Tetris (Grandl et al., 2014) | 50 | 16.4538 | $11.57 \pm 0.60$ | $12.05 \pm 0.80$ | $14.02 \pm 0.52$ | **14.22** |
| Shortest Job First | 100 | 38.3202 | $0.26 \pm 0.03$ | $0.41 \pm 0.04$ | $0.48 \pm 0.02$ | **0.54** |
| Critical Path | 100 | 32.0355 | $8.57 \pm 0.28$ | $8.98 \pm 0.27$ | $9.13 \pm 0.02$ | **9.24** |
| Tetris (Grandl et al., 2014) | 100 | 30.3722 | $13.27 \pm 0.36$ | $12.60 \pm 0.73$ | $14.70 \pm 0.49$ | **15.41** |
| Shortest Job First | 150 | 57.1554 | $0.84 \pm 0.07$ | $1.12 \pm 0.08$ | $1.30 \pm 0.05$ | **1.35** |
| Critical Path | 150 | 48.7963 | $5.33 \pm 0.37$ | $6.27 \pm 0.37$ | $6.65 \pm 0.12$ | **6.85** |
| Tetris (Grandl et al., 2014) | 150 | 44.9376 | $11.21 \pm 0.85$ | $11.44 \pm 0.90$ | $\mathbf{13.04 \pm 0.26}$ | 12.73 |

Table 4: **DAG attack and defense results of Time $\downarrow$ and Ratio (%) $\downarrow$.** The solvers' solutions are recorded and the all the ratio is computed by the solved finish time w.r.t. Normal solution.

| Solver | Mode | job#=50 | | job#=100 | | job#=150 | |
|---|---|---|---|---|---|---|---|
| | | Time$\downarrow$ | Ratio$\downarrow$ | Time$\downarrow$ | Ratio$\downarrow$ | Time$\downarrow$ | Ratio$\downarrow$ |
| Shortest Job First | Normal | 20.9228 | 0.00 | 38.3202 | 0.00 | 57.1554 | 0.00 |
| Shortest Job First | Attack | 21.2093 | 1.37 | 38.5335 | 0.55 | 57.9326 | 1.36 |
| Shortest Job First | Defense | 20.9151 | -0.04 | 38.0470 | -0.71 | 57.4370 | 0.49 |
| Critical Path | Normal | 17.3900 | 0.00 | 32.0355 | 0.00 | 48.7963 | 0.00 |
| Critical Path | Attack | 18.9782 | 9.13 | 34.9976 | 9.25 | 52.1519 | 6.88 |
| Critical Path | Defense | 18.4335 | 6.00 | 33.4258 | 4.34 | 49.9011 | 2.26 |
| Tetris (Grandl et al., 2014) | Normal | 16.4538 | 0.00 | 30.3722 | 0.00 | 44.9376 | 0.00 |
| Tetris (Grandl et al., 2014) | Attack | 18.7944 | 14.22 | 35.0321 | 15.34 | 50.6415 | 12.69 |
| Tetris (Grandl et al., 2014) | Defense | 17.7033 | 7.59 | 34.2604 | 12.80 | 49.2008 | 9.49 |

that the defense RL agent can not only play a defensive role against the attacked problem instance, but can also help further improve the solution of normal instances, as will be shown in some of our experiments. We leave more in-depth analysis and corresponding approach design for future work.

## 4 EXPERIMENTS AND RESULTS

We conduct experiments on three representative tasks: Directed Acyclic Graph Scheduling, Asymmetric Traveling Salesman Problem and Fraud Coverage. The former two problems are popular problems in CO. The third problem is originated from a real-world transaction dataset. The detailed graph embedding for the three tasks is shown in Appendix C. In Appendix G, we provide the training and evaluation parameters of different solvers for fair time comparison and reproducibility. All experiments are run on RTX 2080Ti and RTX 3090 (see Appendix H for the detailed testbed).

### 4.1 TASK I: DIRECTED ACYCLIC GRAPH SCHEDULING

Task scheduling for heterogeneous systems and various jobs is a popular problem due to its practical importance. Many systems formulate the job stages and their dependencies as a Directed Acyclic Graph (DAG) (Saha et al., 2015; Chambers et al., 2010; Zaharia et al., 2012). The data center has limited computing resources to allocate the jobs with different resource requirements. These jobs can run in parallel if all their parent jobs have finished and the required resources are available. Our goal is to **minimize the finish time of the jobs i.e. we should finish all jobs as soon as possible.**

**Solvers.** We choose three popular heuristic solvers as our attack targets. First, the Shortest Job First algorithm chooses the jobs greedily with minimum completion time. Second, the Critical Path algorithm analyzes the bottleneck and finishes the jobs in the critical path sequence. Third, the Tetris (Grandl et al., 2014) scheduling algorithm models the jobs as 2-dimension blocks in the Tetris games according to their finish time and resource requirement.

**Attack model.** The edges in a DAG represent job dependencies, and removing edges will relax the constraints. After removing existing edges in a DAG, it is obvious that the new solution will be equal or better than the original one since there are less restrictions. As a result, in the DAG scheduling tasks, the attack model is to selectively remove existing edges.

**Defense model.** We propose to add non-existing edges on the input graph associated with the CO problem, and obviously the new solution under more constraints is still feasible for the original CO problem. The motivation is to help tune the graph structure to be more suitable for heuristic

algorithms. To reduce the action space, we propose to pre-process the node pairs that already have dependencies and remove the corresponding edges in the candidate set.

**Dataset.** We use the TPC-H dataset (`http://tpc.org/tpch/default5.asp`), which is composed of business-oriented queries and concurrent data modification. Many DAGs have tens or even hundreds of stages with different duration and numbers of parallel tasks. As each DAG in TPC-H dataset represents a computation job, we gather the DAGs randomly and generate three different datasets, TPC-H-50, TPC-H-100, TPC-H-150, of each containing 50 training and 10 testing samples. Each DAG node has two properties: execution time and resource requirement.

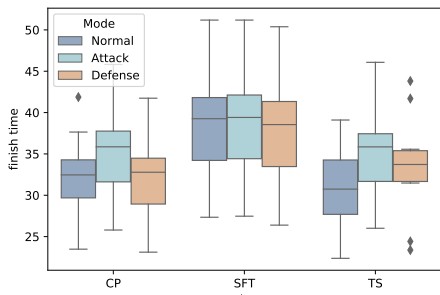

**Results for attack.** Table 3 reports the results of our four attack methods, where RL outperforms other learning-free methods in most cases, illustrating the correctness of our feature extraction techniques and training framework. It is worth noting that even the simplest random attack can cause a significant performance degradation to the CO solvers, showing their vulnerability and the effectiveness of the attack framework.

**Results for attack and defense.** Table 4 and Fig. 3 show the results of attack and defense experiments on DAG. In general, the defense model can compensate for the damage of the attack and can even obtain better solutions than the baseline in some cases. It's also worth noting that for some instances, the edges removed in the attack stage will be added back in the defense.

Figure 3: Finish time ↓ as DAG objective score (mean and std by 10 trials) among three modes: attack, defense and normal: schedule 100 jobs from TPC-H. Attack will incur worse score than in normal mode, which can be remedied by defense.

## 4.2 TASK II: ASYMMETRIC TRAVELING SALESMAN PROBLEM

The classic traveling salesman problem (TSP) is to **find the shortest cycle to travel across all the cities.** Here we tackle the even challenging asymmetric TSP (ATSP) for its generality.

**Solvers.** Four algorithms are treated as our attack targets: i) Nearest Neighbour greedily adds the nearest city to the tour. ii) Furthest Insertion finds the city with the furthest distance to the existing cities in the tour and inserts it. iii) Lin-Kernighan Heuristic (LKH3) (Helsgaun, 2017) is the traditional SOTA TSP solver. iv) Matrix Encoding Networks (MatNet) (Kwon et al., 2021) claims as a SOTA learning-based solver for ATSP and flexible flow shop (FFSP).

**Attack model.** The attack is to choose an edge and half its value, after which we will get a better theoretical optimum. To reduce the action space, we will not select the edges in the current path predicted by the solver at the last time step.

**Defense model.** First we calculate the optimal path by the solver and add these edges to the candidate set. The action is to modify an edge's weight by doubling the distance of that edge in order to encourage the solver to explore other paths.

**Dataset.** It comes from (Kwon et al., 2021) consisting of 'tmat' class ATSP instances which have the triangle inequality and are widely studied by the operation research community (Cirasella et al., 2001). We solve the ATSP of three sizes, 20, 50 and 100 cities. The distance matrix is fully connected and asymmetric, and each dataset consists of 50 training samples and 20 testing samples.

**Results for attack.** Table 5 reports the attack results of four target solvers. In general, the learning-based solvers (e.g. MatNet) or those with intrinsic randomness (e.g. LKH3) show stronger robustness to the attacks. Furthermore, it is notable that the RL based attack outperforms in most cases.

**Results for attack and defense.** Table 6 shows that the defense model works well on ATSP. In addition to making up the degeneration by attack, in some cases it even obtains shorter total distance.

## 4.3 TASK III: FRAUD COVERAGE

Our last problem instance refers to Fraud Coverage (FC), which is an emerging NP-Complete (details in Appendix D.1) problem abstracted from real life: the growing online transactions have also spawned criminals and scams. The transactions $\mathcal{E}$ can be classified into black (fraudulent) events $\mathcal{B}$ and white (normal) events $\mathcal{W}$. To block fraud events, the bank system designs a series of rules $\mathcal{R}$ to

Table 5: **ATSP attack results of Ratio (%)** $\uparrow \pm$ **Std.** Baseline denotes mean tour length on test set. Result is the mean ratio on all test instances computed by the solved tour length w.r.t. baselines.

| Solver | City# | Baseline $\times 10^6$ | Attack Method | | | |
|---|---|---|---|---|---|---|
| | | | Random | OG-Search | SA | RL |
| Nearest Neighbour | 20 | 1.9354 | $10.09 \pm 0.79$ | $9.34 \pm 1.67$ | $10.28 \pm 0.82$ | **12.94** |
| Furthest Insertion | 20 | 1.6092 | $5.35 \pm 0.65$ | $5.18 \pm 0.73$ | $6.78 \pm 0.71$ | **8.56** |
| LKH3 (Helsgaun, 2017) | 20 | 1.4595 | $0.03 \pm 0.02$ | $0.03 \pm 0.03$ | $0.10 \pm 0.07$ | **0.11** |
| MatNet (Kwon et al., 2021) | 20 | 1.4616 | $0.40 \pm 0.08$ | $0.46 \pm 0.04$ | $0.46 \pm 0.06$ | **0.65** |
| Nearest Neighbour | 50 | 2.2247 | $6.24 \pm 0.37$ | $7.02 \pm 0.43$ | $8.14 \pm 0.68$ | **10.26** |
| Furthest Insertion | 50 | 1.9772 | $4.15 \pm 0.36$ | $3.51 \pm 0.63$ | $4.35 \pm 0.45$ | **6.97** |
| LKH3 (Helsgaun, 2017) | 50 | 1.6621 | $0.19 \pm 0.04$ | $0.21 \pm 0.04$ | $\textbf{0.37} \pm \textbf{0.06}$ | 0.35 |
| MatNet (Kwon et al., 2021) | 50 | 1.6915 | $1.39 \pm 0.07$ | $1.71 \pm 0.06$ | $2.01 \pm 0.07$ | **2.15** |
| Nearest Neighbour | 100 | 2.1456 | $4.02 \pm 0.46$ | $3.53 \pm 0.71$ | $3.81 \pm 0.52$ | **5.02** |
| Furthest Insertion | 100 | 1.9209 | $2.88 \pm 0.46$ | $2.97 \pm 0.58$ | $3.35 \pm 0.33$ | **4.87** |
| LKH3 (Helsgaun, 2017) | 100 | 1.5763 | $0.40 \pm 0.04$ | $0.54 \pm 0.03$ | $0.59 \pm 0.02$ | **0.63** |
| MatNet (Kwon et al., 2021) | 100 | 1.6545 | $1.37 \pm 0.06$ | $1.63 \pm 0.03$ | $1.79 \pm 0.04$ | **1.98** |

Table 6: **ATSP attack and defense results of Distance $\downarrow$ and Ratio (%) $\downarrow$.** The solutions are recorded and the ratio is computed by the solved tour length w.r.t. normal solution.

| Solver | Mode | ATSP-20 | | ATSP-50 | | ATSP-100 | |
|---|---|---|---|---|---|---|---|
| | | Distance$\downarrow$ | Ratio$\downarrow$ | Distance$\downarrow$ | Ratio$\downarrow$ | Distance $\downarrow$ | Ratio$\downarrow$ |
| Nearest Neighbour | Normal | 1.9354 | 0.00 | 2.2247 | 0.00 | 2.1456 | 0.00 |
| Nearest Neighbour | Attack | 2.1366 | 10.40 | 2.4264 | 9.07 | 2.2439 | 4.58 |
| Nearest Neighbour | Defense | 1.7564 | -9.25 | 2.2069 | -0.80 | 2.0319 | -5.30 |
| Furthest Insertion | Normal | 1.6092 | 0.00 | 1.9772 | 0.00 | 1.9272 | 0.00 |
| Furthest Insertion | Attack | 1.7088 | 6.19 | 2.0957 | 5.99 | 1.9963 | 3.58 |
| Furthest Insertion | Defense | 1.5210 | -5.48 | 1.9558 | -1.08 | 1.8990 | -1.46 |
| LKH3 (Helsgaun, 2017) | Normal | 1.4595 | 0.00 | 1.6621 | 0.00 | 1.5763 | 0.00 |
| LKH3 (Helsgaun, 2017) | Attack | 1.4598 | 0.02 | 1.6671 | 0.30 | 1.5867 | 0.66 |
| LKH3 (Helsgaun, 2017) | Defense | 1.4595 | 0.00 | 1.6610 | -0.07 | 1.5744 | -0.12 |
| MatNet (Kwon et al., 2021) | Normal | 1.4617 | 0.00 | 1.6915 | 0.00 | 1.6545 | 0.00 |
| MatNet (Kwon et al., 2021) | Attack | 1.4708 | 0.62 | 1.7261 | 2.04 | 1.6841 | 1.79 |
| MatNet (Kwon et al., 2021) | Defense | 1.4591 | -0.18 | 1.6696 | -1.29 | 1.6185 | -2.18 |

identify transactions as either black or white events. The goal is to **select a subset of rules $R \subseteq \mathcal{R}$ to maximize the coverage of fraudulent monetary values while affecting no more than $K$ white events.** The problem can be represented by a bipartite graph, where any edge exists between a rule node and an event node only when the event is deemed as black by the rule. Formally:

$$\max_R \sum_{b \in \mathcal{B}} w(b) \times \mathbb{I}(b \in \bigcup_{r_i \in R} C^+(r_i)) \qquad s.t. \quad | \bigcup_{r_i \in R} C^-(r_i)| \leq K \tag{9}$$

where $w(\cdot)$ denotes the monetary value of a certain transaction event, $C(\cdot)$ denotes the set of events covered by a rule that are deemed as black events, and $C^+(\cdot)$ and $C^-(\cdot)$ denotes the subset of events in $C(\cdot)$ with true labels being black and white, respectively.

**Solvers.** As an emerging real-world CO task, the FC problem is very challenging and here we propose three different solvers as the target for attacking. First, the trivial Local algorithm which iterates over the rules sequentially, adding any rules that will not exceed the threshold. Second, a more intelligent Greedy Average algorithm that always chooses the most cost-effective (the ratio of the increase of black event money values to the increase of number of covered white events) rule at each step until the constraint isn't satisfied. Third, we formulate the problem into standard ILP form (details in Appendix D.2) and solve it by Gurobi.

**Attack model.** Intuitively, when a white event is mislabeled as a black event, the FC problem will achieve an equal or better optimum $f^*(Q')$, since we can possibly cover more white events while not exceeding the threshold. In our attack model, we focus on the attack toward the edges rather than the nodes. We choose to add non-existing black edges that connect rules to black events, which leads to a theoretically better optimum and can potentially mislead the solvers. Further, in order to reduce action space, we only select the unchosen rules, otherwise it will be useless since adding edges for selected rules would not affect a solver's output solution. Here we report the attack method on adding black edges and present the results for attacking black nodes in the Appendix F.1.

**Defense model.** Similar to the attack method, as defense we remove the existing black edges that connect rules to black events. To reduce action space, we select the rules chosen in the prior solution, since deleting black edges for the unchosen rules will not change the solution.

**Dataset.** We analyze the distribution of transaction amounts and rule coverage of the real dataset, then generate a series of simulated data for experiments. The distribution of events amount and the

Table 7: **FC attack results of Ratio (%)** $\uparrow \pm$ **Std.** The Gurobi time limit is shown in brackets w.r.t. different data sizes (it should be long enough to give a feasible solution but not too long for attack). Baseline is the average original solution of the solvers on test set. The ratio here is the mean of ratios on all test instances computed by the solved FC monetary value w.r.t. baselines.

| Solver | problem size: rule#-event# | Baseline | Attack Method (Ratio ± Std) | | | |
|---|---|---|---|---|---|---|
| | | | Random | OG-Search | SA | RL |
| Local Search | 30-3K | 9.5713 | 0.78 ± 0.06 | 0.77 ± 0.11 | 0.85 ± 0.03 | **0.89** |
| Greedy Average | 30-3K | 18.0038 | 2.72 ± 0.16 | 3.17 ± 0.23 | 2.70 ± 0.22 | **4.79** |
| Gurobi(1s) | 30-3K | 18.8934 | 10.41 ± 1.13 | 18.42 ± 1.88 | 18.99 ± 1.95 | **50.68** |
| Local Search | 60-6K | 24.9913 | 0.47 ± 0.04 | **0.80 ± 0.14** | 0.69 ± 0.15 | 0.76 |
| Greedy Average | 60-6K | 43.1625 | 0.91 ± 0.09 | 0.93 ± 0.11 | 1.02 ± 0.09 | **2.29** |
| Gurobi(2s) | 60-6K | 41.1828 | 7.15 ± 0.84 | 9.35 ± 1.02 | 7.02 ± 0.89 | **100.00** |
| Local Search | 100-10K | 22.9359 | 0.76 ± 0.09 | 1.23 ± 0.08 | 0.83 ± 0.09 | **1.55** |
| Greedy Average | 100-10K | 51.3905 | 1.25 ± 0.14 | 1.70 ± 0.34 | 1.37 ± 0.08 | **1.61** |
| Gurobi(5s) | 100-10K | 49.3296 | 6.33 ± 0.70 | 7.69 ± 0.96 | 4.26 ± 0.48 | **92.01** |

Table 8: **FC attack and defense results of Fraud\$** $\uparrow$ **and Ratio (%)** $\uparrow$**.** The solvers' solutions are recorded and the ratio is computed by the solved FC monetary value (Fraud\$) w.r.t. Normal solution.

| Solver | Mode | rule#=30, event#=3K | | rule#=60, event#=6K | | rule#=100,event#=10K | |
|---|---|---|---|---|---|---|---|
| | | Fraud\$↑ | Ratio↑ | Fraud\$↑ | Ratio↑ | Fraud\$↑ | Ratio↑ |
| Local Search | Normal | 9.5713 | 0.00 | 24.9913 | 0.00 | 22.9359 | 0.00 |
| Local Search | Attack | 9.4638 | -1.12 | 24.8038 | -0.75 | 22.6930 | -1.06 |
| Local Search | Defense | 10.0680 | 5.19 | 25.8300 | 3.36 | 23.7252 | 3.44 |
| Greedy Average | Normal | 18.0038 | 0.00 | 43.1625 | 0.00 | 51.3905 | 0.00 |
| Greedy Average | Attack | 17.1256 | -4.88 | 42.3911 | -1.79 | 50.5651 | -1.61 |
| Greedy Average | Defense | 17.6850 | -1.77 | 42.8371 | -0.75 | 51.0684 | -0.63 |
| Gurobi | Normal | 18.8934 | 0.00 | 41.1828 | 0.00 | 49.3296 | 0.00 |
| Gurobi | Attack | 2.7194 | -85.61 | 2.2218 | -94.60 | 4.6731 | -90.53 |
| Gurobi | Defense | 17.2712 | -8.59 | 42.1617 | 2.38 | 51.2941 | 3.98 |

rule coverage is shown in Appendix E. The dataset consists of three rule-event pairs 30-3K, 60-6K and 100-10K, each with 50 training samples and 20 testing samples.

**Results for attack.** Table 7 shows the attack results of our simulated dataset. We can observe that both heuristic and RL approaches have yielded significant attack effects, while RL outperforms the others in most cases (especially for Gurobi, in many cases it is not even possible to give a feasible solution within time after employing RL attacks).

**Results for attack and defense.** Table 8 records the results of attack and defense experiments on FC problems. Experiments are conducted on the same test set. In general, the defender can compensate for the damage of attack effectively and obtain an even better solution than the baseline in some cases. Besides, as a commercial solver, Gurobi should be able to obtain optimal solutions if in sufficient time (assuming we have unlimited computational resources). So we record the time for Gurobi to find the optimal solution under attack and defense. The result is shown in Fig. 4, where Gurobi's solution time after attack (defense) significantly increases (decreases). This inspires us to attack toward the solvers' solution time in future work.

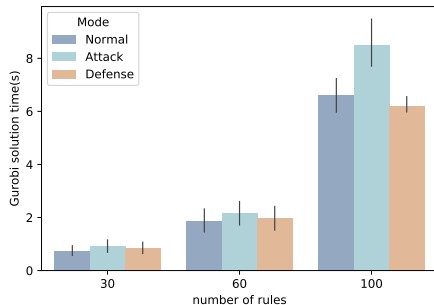

Figure 4: Gurobi's mean time cost in solving FC problems. Run experiments on 3 datasets (20 instances) of different sizes.

## 5 CONCLUSION AND OUTLOOK

We have presented a general adversarial attack and defense framework called ROCO on top of combinatorial solvers. For attack, we devise both RL and traditional heuristic attackers to modify the underlying graph structure of combinatorial problems. Meanwhile, we propose a simple yet effective defense mechanism to modify the ill-posed problem in a reversed way to increase the robustness of combinatorial solvers. Experiments show the effectiveness of our paradigm and techniques.

The proposed paradigm opens up large space for further research, at least in the following aspects: 1) new attack/defense techniques beyond graph structure but also node/edge attribute; 2) iterative adversarial training for defense model, especially for learning-based solvers (at least in the sense of tailored data augmentation); 3) white-box attack/defense when the solver information is known.

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

# APPENDIX

## A  EXAMPLES FOR ATSP AND FC

To have a more intuitive understanding of the attack and defense on ATSP and FC, we provide two examples here. Fig. 5 displays attack and defense effect on Nearest Neighbour algorithm of an ATSP instance. Fig. 6 displays the attack and defense effect on Greedy algorithm of an FC instance with white event threshold 5.

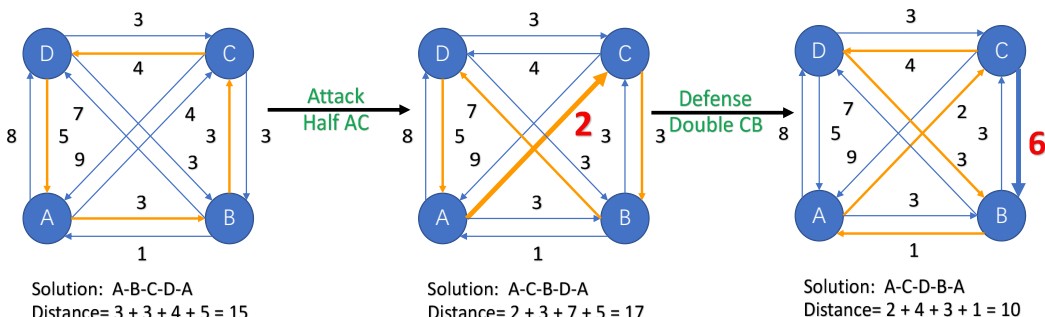

Figure 5: ATSP attack and defense on Nearest Neighbour algorithm. The attack action on edge AC will cause 2 further distance. The defense action on edge CB will help the algorithm improve the solution, even better than the origin.

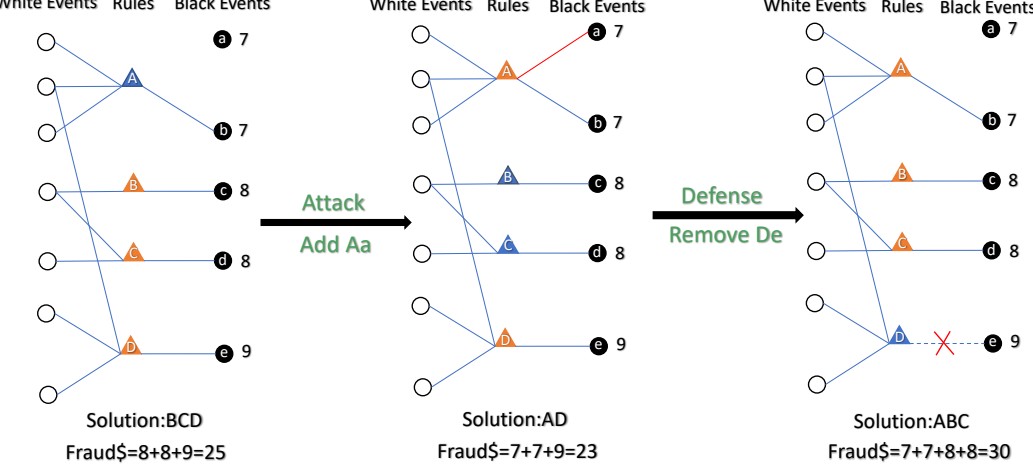

Figure 6: FC attack and defense on Greedy algorithm. The attack action on edge Aa will cause a lower fraudulent monetary value. The defense action on edge De will help the algorithm improve the solution, even better than the origin.

## B    HEURISTIC ATTACK ALGORITHM

As a heuristic attack example, we list the pseudo code of SA in Algorithm 2.

---

**Algorithm 2: Simulated Annealing (SA) Attack**

---

**Input:** Input graphs $\mathcal{G}$; solver $\mathcal{S}$; max number of actions $K$; action sample number $M$;
        Temperature decay $\Delta T$; coefficient $\beta$.

$\mathcal{G}^0 \leftarrow \mathcal{G}; \mathcal{G}^* \leftarrow \mathcal{G}^0; T \leftarrow 1$; # initial temperature

**for** $k \leftarrow 1..K$ **do**
    $flag = False$; # if action is available
    **for** $i \leftarrow 1..M$ **do**
        Random sample an edge $(x, y)$ in edge candidates of $\mathcal{G}^{k-1}$;
        $\mathcal{G}' \leftarrow$ add/delete the edge $(x, y)$ in $\mathcal{G}^{k-1}$; # new state by tentative action
        $P = \exp(\frac{\beta(f(\mathcal{S}|\mathcal{G}') - f(\mathcal{S}|\mathcal{G}^{k-1}) + eps)}{T})$; # action acceptance probability
        **if** $\text{Random}(0, 1) \leq P$ **then**
            flag = True; $\mathcal{G}^k \leftarrow \mathcal{G}'; \mathcal{G}^* \leftarrow \mathcal{G}^k$
            **break**;
    **if** $flag = False$ **then**
        **break**;
    $T = T \cdot \Delta T$;

**Output:** Graph $\mathcal{G}^*$

---

## C    GRAPH EMBEDDING FOR SPECIFIC TASKS

### C.1    TASK I: DIRECTED ACYCLIC GRAPH SCHEDULING

Since the task is a directed acyclic graph, we use GCN to encode the state in the original graph and its reverse graph with inversely directed edges separately. Then we concatenate the two node embedding and use an attention pooling layer to extract the graph-level embedding for Eq. 6:

$$\mathbf{n} = [\text{GCN}_1(\mathcal{G})||\text{GCN}_2(\text{reverse}(\mathcal{G}))], \ \mathbf{g} = \text{AttPool}(\mathbf{n}).$$

### C.2    TASK II: ASYMMETRIC TRAVELING SALESMAN PROBLEM

Considering the graph is fully connected, we use GCN to encode the state in the graph. Then we use an attention pooling layer to extract the graph-level embedding. Eq. 6 becomes:

$$\mathbf{n} = [\text{GCN}(\mathcal{G})], \ \mathbf{g} = \text{AttPool}(\mathbf{n}).$$

### C.3    TASK III: FRAUD COVERAGE

For the RL attack method, different from DAG and TSP, FC has a unique bipartite graph structure. Therefore, we resort to SAGEConv, which can handle bipartite data, for graph feature extraction. As input, we classify the nodes into three classes (rules, black events and white events) and associate them with three dimension one-hot tensors. Besides, we add one more dimension for event nodes, which records their amounts. Eq. 6 becomes:

$$\mathbf{n}_e = \text{SAGEConv}_1(I_r, I_e), \ \mathbf{n}_r = \text{SAGEConv}_2(I_e, I_r)$$
$$\mathbf{g}_e = \text{AttPool}_1(\mathbf{n}_e), \ \mathbf{g}_r = \text{AttPool}_2(\mathbf{n}_r) \tag{10}$$

# D  FORMULAS AND PROOFS

## D.1  FC NP-COMPLETE PROVEMENT

For the sake of our proof, here we redefine FRAUD-COVERAGE and give the definition of an existing NPC problem: SET-COVER.

FRAUD-COVERAGE. Given a set of rules $\mathcal{R}$ along with an event set consisted of white and black events $\mathcal{E} = \mathcal{W} \cup \mathcal{B}$. Rules are corresponding to certain events (white $W(r)$ or black $B(r)$) which have their amounts $w(e)$. Does there exist a collection of these rules to cover $\geq M$ fraudulent monetary value while influence no more than $k$ white events?

SET-COVER. Given a set $U$ of elements, a collection $S_1, S_2, \ldots, S_m$ of subsets of $U$, and an integer $k$, does there exist a collection of $\leq k$ of these sets whose union is equal to $U$?

First, we need to show that FC is NP. Given a set of selected rules $R = \{r_1, r_2, \ldots, r_m\}$, we could simply traverse the set, recording the covered black and white events. Then, we can assume whether the covered black events number is no more than $k$ and fraudulent monetary value is no less than $m$.

Since the certification process could be done in $O(n^2)$, we could tell that FC is NP. Then, for NP-hardness, we could get it reduced from SET-COVER.

Suppose we have a SET-COVER instance, we construct an equivalent FC problem as follows:

- Create $|U|$ black events, each with amount 1.
- Create $m$ rules, set $B(r_i) = S_i$.
- Connect each rule to a different white event of amount 1.
- Set the white events threshold $k_w$ equal to the set number threshold $k_s$.
- Set fraudulent monetary target $m = |U|$.

Suppose we find a set of rules which meet the conditions of FC, then we select subsets $S_i$ iff we select $r_i$. The total number of rules (subsets) is no more than $k_w(k_s)$ since the influenced white events number is equal to $|R|$. The subsets also cover $U$ since the covered white event money (each with amount 1) is no less than $|U|$. Similarly, we can prove that we can find a suitable rule set for FC if we have found a set of subsets that meet the conditions of SET-COVERAGE. So we can induce that SET-COVER $\leq_p$ FRAUD-COVERAGE.

Thus, we prove that FC is NP-Complete.

## D.2  FC ILP FORMULATION

As discussed in the main text, $\mathcal{B}$ denotes the set of black events, $\mathcal{W}$ represents the set of white events, $\mathcal{R}$ refers to the rule set and $\mathcal{E}$ denotes the event set. Using notations above, we could translate Eq. 9 into standard ILP form as follows:

$$\max \quad \sum_{i \in \mathcal{B}} Y[i] \times W[i], \quad s.t. \quad \sum_{i \in \mathcal{W}} Y[i] \leq K$$
$$\text{for } i = 1 \ldots |\mathcal{E}|, \quad (Y[i] - 0.5)(0.5 - \sum_{j=1}^{|\mathcal{R}|} X[j] \times \mathbb{I}(i \text{ in } E[j])) \leq 0 \tag{11}$$

where $X[j] \in \{0, 1\}$ denotes whether rule $j$ is chosen or not, $Y[i] \in \{0, 1\}$ shows whether event $i$ has been covered by the chosen rules. Besides, $W[i] \in \mathbb{R}$ records the amount of the events while $E[j] \subseteq \mathcal{E}$ is the corresponding events of rule $j$. The third equation ensures the event binary to be 1 iff it has been covered by the rule set (if $\exists X[j] = 1$ and event $i \in E[j]$, then the formula in the second bracket is negative, ensuring $Y[i]$ to be 1; else if event $i$ is not covered by any chosen rules, then the second formula is positive and $Y[i]$ must be 0).

# E  FRAUD COVERAGE DATASET

The Fraud Coverage dataset is generated according to the real data. We normalize the amount of each event to $[0, 1]$. Fig. 7 shows the distribution of the amount larger than $0.1$ of a rule100-event10K instance. Most events are small transactions and show a long-tail distribution format while fraud events tend to obey the long-tail distribution and are close to the uniform distribution when the amounts are larger than $0.1$. Fraudsters often tend to cheating larger amounts and disregarding the small transactions may account for this phenomenon. Fig. 8 shows fraudulent monetary value coverage of different strategies. It follows the real condition that few well-designed rules can cover most fraud events. The other rules can be regarded as complementary to the main rules. Different rules do not show some obvious patterns on the distributed number of events. For example, the fraudulent monetary value detected is not linearly related to the distributed events. Different rules may also cover the same fraud events and normal events, which makes the problem complex (Appendix D.1). That's why we need to select a suitable rule set.

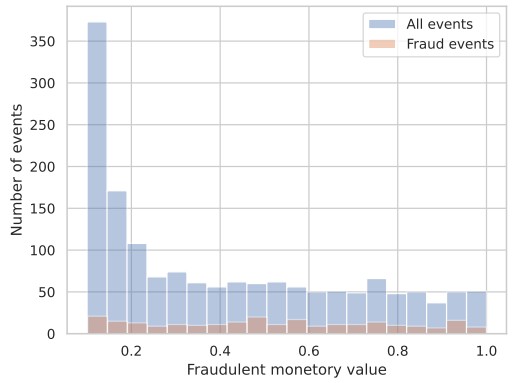
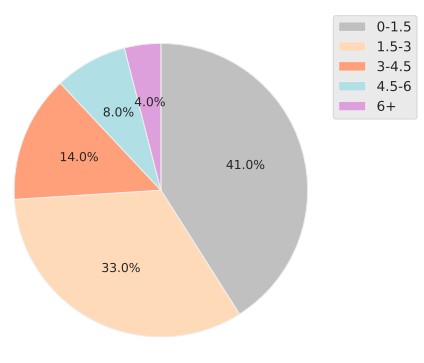

Figure 7: FC dataset fraudulent monetory value distribution of a 100-10K instance.

Figure 8: The distribution of fraudulent monetary value covered by different rules. The $x - y$ pair means the rules can detect amount in range $[x, y]$.

# F  ADDITIONAL EXPERIMENT RESULTS

## F.1  TABLES

As discussed in Sec. 4.3, we propose 2 attack methods towards the FC solvers. The experiment results of 'adding black edges' are shown in the main text. And the results of the W2B attack method are shown in Table 9.

Table 9: **FC Node attack results of Ratio (%)↑.** Baseline is the average original solution of the solvers on test set. We use RL as our attack method and the ratio here is the mean of ratios on all test instances computed by the solved FC monetary value w.r.t. baselines.

| Solver | Rule#-Event# (Baseline, Ratio) | | |
|---|---|---|---|
| | 30-3K | 60-6K | 100-10K |
| Local Search | 9.5713, 0.54% | 24.9913, 1.15% | 22.9359, 0.75% |
| Greedy Average | 18.0038, 0.11% | 43.1625, 0.27% | 51.3905, 0.18% |
| Gurobi | 18.8934, 41.47% | 41.1828, 78.59% | 49.3296, 69.84% |

## F.2  FIGURES

Fig. 9 and 10 summarize the attack and defense results of TSP and FC experiments.

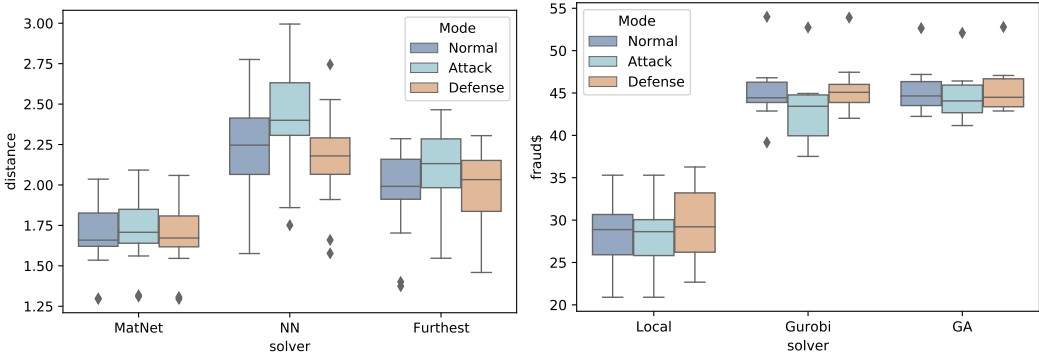

Figure 9: Box diagram for TSP attack and defense results. We run experiments on 20 testing instances which traverse 50 cities from 'tmat'. Three solvers are chosen as our targets and their solution distance are shown in the figure.

Figure 10: Box diagram for FC attack and defense results. We run experiments on 20 testing instances that contain 60 rules and 6000 events. Three solvers are chosen as our targets and their solution fraud$ are shown in the figure.

## G  EXPERIMENT PARAMETERS

**Reinforcement learning model settings.** Table 10 records the parameters for RL during the training process. Trust region clip factor is a parameter in PPO agent to avoid model collapse. We also adopt some common policy-gradient training tricks like reward normalization and entropy regularization during training processes.

Table 10: RL parameter configuration in tasks GED, ATSP and FC

| Parameters | DAG | ATSP | FC |
|---|---|---|---|
| Actions# | 20 | 20 | 10 |
| Reward discount factor | 0.95 | 0.95 | 0.95 |
| Trust region clip factor | 0.1 | 0.1 | 0.1 |
| GNN type | GCN | GCN | SAGEConv |
| GNN layers# | 5 | 3 | 3 |
| Learning rate | 1e-4 | 1e-3 | 1e-3 |
| Node feature dimensions# | 64 | 20 | 16 |

**Attackers evaluation setting.** For fair comparison of different attackers Random, OG-search, SA and RL, we set the parameters to ensure similar evaluation time. According to the time complexity we discuss in Table 2, we specify the following parameters: number of iterations $N$, beam search size $B$ and number of different actions $M$ in each iteration.

**DAG** : Random $N = 30$; OG-search $B = 3$, $M = 9$; SA $N = 5$, $M = 6$; RL $B = 3$, $M = 9$;

**TSP** : Random $N = 130$; OG-search $B = 5$, $M = 25$; SA $N = 13$, $M = 10$; RL $B = 5$, $M = 25$;

**FC** : Random $N = 220$; OG-search $B = 6$, $M = 36$; SA $N = 22$, $M = 10$; RL $B = 6$, $M = 36$;

## H  EXPERIMENT ENVIRONMENTS.

DAG and TSP experiments are run on GeForce RTX 2080Ti (11GB) and Intel(R) Core(TM) i7-7820X CPU @ 3.60GHz. FC experiments are run on GeForce RTX 3090 (20GB) and AMD Ryzen Threadripper 3970X 32-Core Processor. Our environment configurations are as follows:

- Ubuntu 20.04
- CUDA 11.2
- Pyhton 3.7
- Pytorch 1.9.0
- Pytorch Geometric 1.7.2

