# OpenReview forum: "Mind Your Solver! On Adversarial Attack and Defense for Combinatorial Optimization"
_ICLR.cc/2022/Conference — ICLR 2022 Submitted_

### Official Review · Reviewer_pi8H · 2021-10-28

**Correctness:** 1
**Technical Novelty And Significance:** 2
**Empirical Novelty And Significance:** 3
**Recommendation:** 3
**Confidence:** 3

**Main Review:**

$\textbf{Strengths}$
-	Enhancing the robustness of combinatorial optimization solver is an interesting and timely problem.
-	Under the proposed scheme, the presented attack and defense methods are reasonable.

$\textbf{Weaknesses}$
-	The main weakness is that the considered attack-defense scheme seems to be problematic. From the very beginning, I am confused about the definition of an attack. In my opinion, an attack on the CO-solver should try to prevent the CO-solver from producing a high-quality solution to the *original input* (but not the perturbed input). But the proposed attack method aims to maximize $f(S(Q^{‘}|G^{‘}))$ rather than $f(S(Q^{‘}|G))$.
-	Similarly, the effect of the defense should be measured by the quality of the solution to the original input, which is however unknown to the defender because it has been perturbed by the attacker. In such a sense, if the defender knows the attacker’s strategy, they can mitigate the attack by identifying the removed edge; otherwise, unless an input distribution is known, there seems no meaningful defense method.
-	Assuming the proposed attack-defense scheme is reasonable, the proposed reinforcement methods are reasonable but do not contribute new techniques.
-	In experiments, maybe I missed something, but I did not see the setting of the number (K) of edges that are allowed to be changed; to me, experimenting with different K is necessary. In addition, only three fixed graphs are considered for DAG scheduling, and for demonstrating its statistical significance, it is better to experiment with a collection of random instances and examine the average performance.

It would be better if the paper could elaborate on the following questions:
- Why strong nonlinearity and NP-hardness can cause the vulnerability defined in this paper?
- What does it mean by saying that the parameters are universal?
- From the attacker’s standpoint, what is the purpose of setting $f(Q^{‘})\geq f^*(Q)$? I do not see that such a setting can result in a better attack.
- Relaxing the candidate space does not necessarily mean that the solver should perform better, because the performance is measured with respect to the optimal solution which can have a better objective value in the relaxed space. For example, if we relax the metric space to general space, some problems can exhibit stronger approximation hardness.



**Summary Of The Paper:**

This paper presents a framework of adversarial attack and defense for solvers to combinatorial optimization problems. In particular, the attacker can modify the problem instance by removing graph edges, while the defense side seeks to add edges to mitigate the attacking effect. The paper presents attack methods based on reinforcement learning and heuristic methods, and proposes a defense method based on reinforcement learning. Experimental studies are provided on three combinatorial problems.

**Summary Of The Review:**

This paper considers an interesting problem, but the adopted settings need better justifications, and the proposed methods are unfortunately not novel.

---

> ### Author Response · Authors · 2021-11-10
> **Response to review.**
>
> Thank you for your time and comments. And our responses are as follow:
> - For attack, suppose we loosen part of the constraints of a CO problem, then the feasible solution space will be expanded and we will get a better (at least the same) theoretical optimal solution to the new problem (and we will expect our solver to get a better solution to the new problem). However, some CO solvers may not adapt to this kind of revision and will get an even worse solution than the original one, and that is what we called non-robustness.
> Here we focus on finding the non-robustness of the solver in the generated new instance. And the original solution $f(S(Q)|G)$ is one feasible solution under harder constraints $Q$, which is also feasible in the new problem as a baseline.  And $f(S(Q')|G')$ is the solution to the new problem, which we expect to be better than the baseline (since the optimal value gets better), but gets worse after attack.
> - Actually, our defense model is a general one and you can regard it as an optimizer for any CO problems, either the original problem or the problem after attack. We don't need to know what attack was implemented in it and only need to modify the graph to obtain a better result. Although the inverse actions of the attack is a type of defense, the defense models can train their own strategy to operate on e.g. different edges.
> Specifically, for defense, we adopt the similar idea to attack. We add additional constraints to the original problem, thus the new solution (under stronger constraints) is absolutely feasible for the original one. In this way, we keep modifying the graph structure (i.e. constraints) and running the solver again to get a new solution, adapting it to the original problem to get a better solution value (although the optimal solution could not be better than the original one, the solver solution could get better under an easy-to-solve graph structure).
> - For the novelty. The main contribution is that we propose an adversarial framework to help detect the non-robustness of CO solvers during attack and improve the solvers' performance by finding a better initial state during defense.  Our framework is flexible and versatile: 1) For attackers, they can be algorithms such as random, heuristic algorithms as well as RL. 2) For solvers, they can be any CO solvers that could be treated as a black box.
> - The setting of the number(K) of edges is shown in Appendix G. The parameter is quite important as you point out and we will emphasize it in the experiments in the revised version. The suggested experiment of different K and the different sizes of the graph is also considerable.
>
> Then, we can discuss the questions that could be elaborated on:
> 1. The discrete solution and the decision sequence may account for the phenomenon. The small perturbation of the solution space will result in a huge change in the decision sequence. In the paper, our work mainly focuses on experimental validation. And theoretical analysis is one of our considerable future works.
> 2. This question confuses me a little and we don't say that the parameters are universal. Try to understand your concern, I find the "universal" in the paper. In the paper, we only mention that our framework is universal across tasks and solvers. However, the parameter settings for different solvers are not universal and they are listed in the appendix.
> 3. Actually, the purpose of setting $f(Q') \geq f(Q)$ is not to find a better attack. The CO problems are usually NP-hard and we can not find the optimal solution in the limited time. We construct $Q'$ from $Q$ by loosening the constraints and this process will ensure $f^*(Q') \geq f^*(Q)$ so that the feasible solutions of $Q$ will be a subset of $Q'$.  The problems become simpler while the solver will find the unreasonable worse answer, and that is what we called non-robustness.
> 4. Yes, the actions space of NP-hard problems is hard to analyze. As we discuss in 3, the problems become simpler as the constraints become less, feasible solution space gets larger and the optimal solution will become no worse. However, we actually do not care about finding the optimal solution but only a good solution. While the better solution space becomes larger and easier to find, the solver inversely gives unreasonable or even worse solutions, that is what we called non-robustness.
> **You could also refer to our official comments for more details.** We sincerely hope that our explanations can answer your questions and help you understand our work better. We are looking forward to further discussions to discover more pros and cons of this paper.

---

### Official Review · Reviewer_CZdE · 2021-11-01

**Correctness:** 2
**Technical Novelty And Significance:** 1
**Empirical Novelty And Significance:** 2
**Recommendation:** 3
**Confidence:** 4

**Main Review:**

Concerns:
- The defined *robustness*, *adversarial attack* and *defense* in this paper are not reasonable. The paper considers the differences between $f(S(Q^\prime)|G^\prime)$ and $f(S(Q)|G)$ as the robustness of a solver, which makes little sense. It is common that a small perturbation in the graph can make the optimal result of a CO problem change a lot, but this is an intrinsic property of CO problem, and is unrelated to the *robustness* of a solver. A reasonable definition of *robustness* can be $-(f(S(Q)|G)-f^*)$, where $f^*$ is the ground truth optimal value. Then the attack is to find some $G$ that makes some solvers find sub-optimal solutions that are far from the ground truth optimum.
- For the *defense*,
    - What does "the defense RL agent can not only play a defensive role against the attacked problem instance, but can also help further improve the solution of normal instances" mean? I think modifying the graph itself to get better results is unacceptable for CO tasks. After modification, the problem becomes a different problem, and the solution to it means little to the original problem.
    - For Eq.4, how can the defender access the original graph $G$ to be part of its objective? If the original graph is accessible, then no other actions are meaningful as we should directly adopt the original graph.
- The scope of the claims in this paper are too broad and should be narrowed down to specific tasks and solvers that the proposed ROCO can work on. For a trivial example, the shortest path finding problem is a combinatorial optimization problem with deterministic, optimal and efficient solver. The Bellman Ford method or Dijkstra method for it are deterministically robust and should not be able to be attacked in any ways.
- The proposed solutions have limited novelty.

Minor:
- Should it be $f(S(Q^\prime)|G^\prime) -f(S(Q^\prime| G))$ in Eq.4?

**Summary Of The Paper:**

This paper studies the adversarial attacks and defenses problem of combinatorial optimization problems. The paper presents ROCO framework, to conduct attack and defense by maximizing/minimizing the difference between the calculated optimum values between the perturbed graph and the original graph. Both RL-based and heuristics-based solutions are proposed and implemented on three combinatorial optimization tasks.

**Summary Of The Review:**

My main concern is about the problem setting. I think the definition of *robustness*, *adversarial attack* and *defense* about CO in this paper are unreasonable. By modifying the original graph in CO tasks, the problem becomes a different problem and the solution lose its meaning to the original problem.

---

> ### Author Response · Authors · 2021-11-10
> **Response to Review.**
>
> Thank you for your time and comments. And our responses are as follow:
> - For reasonability of ROCO, it's true that we could find some adversarial $G$ that yield sub-optimal solutions (away from the ground truth optimal value). However, it's not practical to get the ground truth optimal value for CO problems due to their NP-hardness. Hence, we bypass this and come up with another comparison method to reveal the solvers' vulnerability. For attack, suppose we loosen part of the constraints of a CO problem, then the feasible solution space will be expanded and we will get a better (at least the same) theoretical optimal solution to the new problem (and we will expect our solver to get a better solution to the new problem). However, some CO solvers may not adapt to this kind of revisions (with a better optimal solution) and will get an even worse solution, and that is what we called non-robustness.
> - It's our negligence that we haven't explained our notifications clearly and caused your misunderstanding of our defense model. In Eq.2, $G$ is the original graph, but in Eq.4 $G$ represents the graph after the attack (so the defender could not access the original graph).  Opposite to the attack model, we add constraints during defense, ensuring that the solution of the new problem is also feasible to the attacked one. Then we adapt the solution to the attacked problem to get a better solution (although the optimal solution could not be better than the one after attack, the solver solution could get better under an easy-to-solve graph structure).
> - Same as the first point, the paper focuses on the NP-hard CO problems. Your given example, problems like shortest path finding which already have polynomial solvers are not included in our research scope. The NP-hardness leads to our inability to find the optimal solution to the problem and is the direct cause of the way we attack.
> - As we've explained above, $f(S(Q')|G)$ in Eq.4 means we adapt the solution of the new problem to the attacked one to get a better solution value.
> **You could also refer to our official comments for more details.** We sincerely hope that our explanations can answer your questions and help you understand our work better. We are looking forward to further discussions to discover more pros and cons of this paper.

---

### Official Review · Reviewer_Pdsc · 2021-11-02

**Correctness:** 1
**Technical Novelty And Significance:** 3
**Empirical Novelty And Significance:** 1
**Recommendation:** 1
**Confidence:** 5

**Main Review:**

Overall, I like the challenge that authors aim to tackle a lot. The problem is highly important and studying robustness of comb. opt. solvers is highly important. Moreover, the paper is easy follow (even though, the notation is rather confusing; see below).

However, there are some major problems with the paper:

1: The main problem is indeed the actual problem formulation of Eq. (2). This equation is not helpful in studying the robustness of the solvers. Note that the underlying graph G' (resp. Q') is different -- thus, also the objective function of the optimization problem when feeding in this new input *should likely change*. Put differently: Eq. (2) is merely identifying two problem instances G and G' which lead to different objective function values. It does not tell anything about the robustness of the solvers.

[Side note: One might argue that the authors aim for small modifications to G and, thus, the objective function value should also change only slightly. However, in particular for NP-hard comb. opt. problems such assumption is not correct. Indeed, small changes in the input can lead to larger changes in the output. In this case, this is not an issue of the non-robustness of the solver but inherent to the problem setting.]

Given this, all follow up experiments and insights are, unfortunately, not helpful.

If we want to study robustness of the solvers, it would be more appropriate to analyze whether the prediction of the solver is correct when feeding in G -- but incorrect when feeding in G' (in whatever sense correctness is measured). In the current formulation, though, this is not captured at all.

2: The definitions/notations are rather confusing: Q is used as a problem defined on a graph G. What exactly is Q? It seems to be fully determined by G (see Eq. 2). But why is the solver then operating on Q and not G? And what exactly does then the conditioning on G mean?

Moreover, if Q is fully determined via G (resp Q' via G'), how does Eq. 4 make sense? Here the solver is applied on Q' but conditioned on G. How is this possible?

Finally, in some cases f is operating on S(Q) -- at other places f is directly operating on Q, i.e. f(Q). I would appreciate to fix these notations.

3: Experiments: As mentioned above, I am convinced that the current experiments are not helpful in investigating the robustness of solvers as claimed by the authors.

Besides that, I found the attack to the TSP unclear. The authors state that they "choose an edge and half its value". However, in the model definition they only define edge addition and deletion. Either the authors should update their model description or perform a different kind of experiment.

4: The paper strongly relates their motivation, model, and experiments to graph attacks/defenses. However, the related work on this is very weak. There is a huge literature on these topics which should be discussed more carefully. In particular, I was wondering, why a simple gradient -based attack (relaxing the discrete nature of the data) was not used as comparison.

5: Regarding technical novelty: The authors are highly inspired by other graph based (RL) attacks. Thus, the novelty in this regard is limited as well.


**Summary Of The Paper:**

The authors aim to study the robustness of combinatorial optimization solvers (in particular ones that are based on learning). They treat the task as a graph problem and perform attacks on the underlying graph structure. Additionally, they aim to propose a defense mechanism. Finally, experiments on three different tasks are performed.

**Summary Of The Review:**

The paper aims to tackle an important problem. However, the introduced model definition (Eq. 2) is incorrect / not helpful in this regard. Thus, unfortunately, the conclusions drawn are very questionable. Moreover, notations, related work, and experiments needs improvements.

---

> ### Author Response · Authors · 2021-11-10
> **Response to Review.**
>
> Thank you for your time and comments. Please see our responses as follow:
> - For the rationality of problem formulation, $G'$ is obtained from $G$ by loosening some specific constraints in $G$ (but not random small modifications). It ensures $G'$ has a no worse theoretical optimal solution than $G$ and becomes a easier problem. For example, in DAG problem, $G'$ is obtained by removing some dependency of task relationship in $G$. Specifically, in our setting, a successful attack is defined as: Loosen part of the constraints of the original CO problem, re-run the solver on the new problem but get a worse solution. The attack could reveal the vulnerability of the solver for that the solver could not adapt to the new scenario (with a better optimal solution) and give an unreasonable worse solution.
> - It's our negligence that we haven't explained our formulation well in the paper. Here we do some compensated explanations to understand our settings better. In Eq(1), we first introduce $f(x|G)$, where $x$ is a solution to the problem and $f$ is the value function (mapping a solution to a value) of a certain CO problem. Besides, as we've described in the paper, $G$ is the graph that encodes the problem constraints (e.g. The DAG problem could be encoded by a directed graph), and $f(x|G)$ means we calculate the solution value of $x$ on graph $G$ (a bit different from "conditional on"). We emphasize "on graph $G$" here because we may use the solution on different graphs. For example, we use $f(S(Q')|G')$ in attack to compare the solver's performance in different problem settings, but use $f(S(Q')|G)$ during defense to get a better solution value in the original problem setting. Finally, $Q$ stands for a specific CO problem we wish to solve and $x=S(Q)$ denotes the solution we get by the solver $S$.
> - At the bottom of page 3 of the paper, we give the definition of the attack "In this paper, concretely our attacker g is allowed to modify edges (e.g. adding or removing edges) from G to construct the new graph." Here we take adding and removing edges as examples of modifying edges while halfing and doubling edge values are also allowed.
> - In this work, we treat the solvers as black boxes and use RL and other heuristic methods to conduct attack. It is worth mentioning that most of the attacked solvers are not learnable (such as the SFT algorithm) and cannot give gradients for gradient-based attack. But for learnable solvers (such as MatNet), we could treat them as white boxes and conduct gradient-based attacks. It's an interesting idea and we may treat it as our future work.
> - Above all, this paper's main contribution is that we propose an adversarial framework to help detect the non-robustness of CO solvers during attack and improve the solvers' performance by finding a better initial state during defense.
> **You could also refer to our official comments for more details.** We sincerely hope that our explanations can answer your questions and help you understand our work better. We are looking forward to further discussions to discover more pros and cons of this paper.

---

> > ### Comment · Reviewer_Pdsc · 2021-11-26
> > **Thank you**
> >
> > Thank you for your feedback.
> >
> > 1) I am still not very convinced about the set-up. Indeed, it could also be that the solution obtained after the "attack" is actually very close to the (NP-hard to find) optimal solution. Or put differently: That the prediction on the original (unperturbed) problems is simply wrong -- and that the increase in the objective function score is actually reasonable. A statement like "that the solver could not adapt to the new scenario" is, thus, a bit critical.
> >
> > Moreover, I still feel that the basic idea and motivation is not well explained in the paper (which is also due to the confusing notation).
> >
> > 2) Notation: As said, I still feel that this is confusing. On one hand you are saying x = S(Q) to indicate the output x of the solver S on the problem Q. How is it then possible/reasonable to have Q' and G in the same function? Why should we compute "the value of the solution for Q'" using graph G? This value has not meaning. (Note also again that in particular for NP-hard combinatorial opt. problems, the solutions for Q and Q' might vastly differ by design.)
> >
> > 3) The RL attack described on page 4 clearly states "Action. As mentioned in Sec. 3.1, the attacker is allowed to add/delete edges in the graph". The TSP attack, however, is using a different principle.
> >
> > Overall, I like the idea of the paper and feel it is a very important task. However, the definitions, execution of experiments, and clarity need to be improved significantly before acceptance. I strongly encourage the authors to revise the paper.
> >
> > My overall score at the moment would be still between 1-3.

---

> > > ### Author Response · Authors · 2021-11-27
> > > **Response to review.**
> > >
> > > Thank you for your approval of our paper's idea and your critical insights. We have clearly recognized the need to revise our paper and our responses to your confusion are as follows:
> > >
> > > 1. To have a more clear impression of our "attack", you could refer to the following schematic:
> > >
> > >      --------------------  Attacked Feasible Solution
> > >
> > >      --------------------  Original Feasible Solution
> > >
> > >      --------------------  Original Optimal Solution
> > >
> > >      -------------------- Attacked Optimal Solution
> > >
> > >      where a higher line denotes a larger solution value (we want the smaller the better). A bit confused about your "simply wrong", we claim that all the solvers being attacked in our paper can recognize the constraints and always give feasible solutions. Under this setting, we perturb the original problem by relaxing some constraints, which will yield a better (smaller) optimal solution. And we say that an attack successes if the solver gives a larger feasible solution than before (we hope that the new feasible solution should be smaller than the original one). As shown in the schematic, the gap between "feasible" and "optimal" becomes bigger after being attacked, and that is, what we say, "the solver cannot adapt to the new scenario". Since the optimal solution is NP-hard to find, we use this kind of definition (ensuring the gap becomes larger) for our attack.
> > >
> > > 2. First, we state that f(Q) is a feasible solution of problem Q given by the solver (e.g. in DAG f(Q) is a permutation of the tasks following the sequential constraints). Then for our defense, the defender will perturb the original problem by adding some constraints, which ensures that the new solution is still feasible for the original problem. For example, in DAG if we add a directed edge (see Figure 2 in our paper) in the graph and get a new solution (permutation of the tasks) by the solver, the solution is still feasible for the original problem which we haven't added the edge. Recall that our target for defense is to get a better solution for the original problem (different from attack, which targets at getting a larger gap), so we put the new solution back in the original graph to get a potentially better solution value. It could be confusing due to our unclear notations and we will revise them thoroughly in the new version.
> > > 3. That's really our negligence in the expression. At the bottom of page 3, we said "In this paper, concretely our attacker g is allowed to modify edges (e.g. adding or removing edges) from G to construct the new graph." The attacker is allowed to conduct any kind of modifications on the edges such as add/delete or modify the edge weights, and we only use add/delete as a concise example.

---

> > > > ### Comment · Reviewer_Pdsc · 2021-11-27
> > > > **Reply**
> > > >
> > > > Thanks for your feedback and thanks for clarifying that you assume always feasible solutions (I think you have to highlight this more clearly in the paper because it is an essential point!).
> > > >
> > > > Your figure helps a lot to explain my point, actually. You are trying to "push" the line
> > > > -------------------- Attacked Feasible Solution
> > > > as high as possible compared to
> > > > -------------------- Original Feasible Solution
> > > >
> > > > What would be more important, however, is the gap between
> > > > -------------------- Attacked Feasible Solution
> > > > and
> > > > -------------------- Attacked Optimal Solution
> > > >
> > > > I agree with you that this gap (assuming Eq. 2 becomes positive) is larger than the gap between "original feasible" and "original optimal". However, since (i) we don't know the original gap, and (ii) the objective function value of such NP-hard problems might change dramatically, it is rather difficult to interpret whether an "increased gap" is really critical.
> > > >
> > > > Moreover, even if Eq. 2 becomes negative (i.e. a seemingly non-succesfull attack) the gap after the attack might be larger.
> > > >
> > > > Thus, you actually don't need the line
> > > > -------------------- Original Feasible Solution
> > > > at all. In Eq. (2) you can simply remove the second term f(S(Q)|G) since it anyways does not depend on G'.
> > > >
> > > >
> > > > Regarding my "simply wrong" wording: What I meant is that the original gap might be already very large. Thus, the solver is simply wrong in its prediction.

---

> > > > > ### Author Response · Authors · 2021-11-27
> > > > > **Reply.**
> > > > >
> > > > > Thanks for your timely reply.
> > > > >
> > > > > I agree with the point that $f(S(Q)|G)$ is redundant in our Eq. (2). Actually, I simply write this item to indicate that we are trying to push the attacked feasible solution to be higher than the original one. Since the optimal solution is NP-hard to find, we push Eq(2) to be positive to ensure that the attacked gap is larger than the original gap.
> > > > >
> > > > > Second, for your concern that the "increased gap" may not be really critical, I think we could put our sights on solvers' robustness. The attacked methods designed in this paper (e.g. RL) are trying to find adversarial instances that the solvers' solution is far from the optimal one, revealing the solvers' non-robustness. Besides, to overcome this non-robustness, we develop defense methods in our paper to find better solutions for the CO problems. So our ultimate motivation is to reveal current solvers' non-robustness and to develop more robust CO solvers. It is important since CO problems are universal and their conditions may evolve over time (we can really encounter adversarial instances in real-life).
> > > > >
> > > > > The question is that the "gap" cannot be quantified since the optimal solution is NP-hard to find, but we believe it can reveal the solvers' sensitivity and drive us to develop more robust solvers which can adapt to more scenarios (e.g. using the defense method in our paper).

---

> > > > > > ### Comment · Reviewer_Pdsc · 2021-11-27
> > > > > > **One more reply**
> > > > > >
> > > > > > I think we have the same understanding that solvers' robustness is essential. No discussion required here.
> > > > > >
> > > > > > Let me be a bit more careful in my explanation what I meant with the "increased gap" above: You are saying "The attacked methods designed in this paper (e.g. RL) are trying to find adversarial instances that the solvers' solution is far from the optimal one". Correct, this would be the goal. However, we do not know how large this gap is -- we can only measure the difference to the feasible original solution. What does this tell us?
> > > > > >
> > > > > > Simple example
> > > > > >
> > > > > > original feasible: 100
> > > > > >
> > > > > > original optimal (unknown): 20
> > > > > >
> > > > > > attacked feasible: 110
> > > > > >
> > > > > > attacked optimal (unknown): 19 (or alternatively, since it could also dramatically change, drop to 1)
> > > > > >
> > > > > > What do we learn from the difference 110-100 you are looking at? I think it would be interesting to discuss this in the paper.

---

> > > > > > > ### Author Response · Authors · 2021-11-27
> > > > > > > **Reply.**
> > > > > > >
> > > > > > > Thanks for your reply.
> > > > > > >
> > > > > > > Currently in my opinion, maybe we cannot impose meaning on the difference 110-100. As you've claimed before, the problem after the attack has been changed and the difference only makes sense under the setting "the optimal solution becomes better" since it ensures that the gap becomes larger. However, I think it's really an interesting problem and we will dig it deeper.
> > > > > > >
> > > > > > > Then for our "gap", I think we can take our Gurobi as a typical example. For a particular solver, **a larger gap means a harder problem for it to solve**. Gurobi is a commercial solver and it can find the optimal solution to CO problems in our paper under unlimited resources. However, as shown in Figure 4, the solving times for Gurobi get larger after being attacked (in this figure we run Gurobi to find the optimal solutions and record their solving times). Recall that we just relax some constraints to get the new problem, but Gurobi's solving time becomes larger, which indicates that we have discovered the vulnerability inside Gurobi and found some harder problem instances for it.

---

### Official Review · Reviewer_SYfi · 2021-11-03

**Correctness:** 2
**Technical Novelty And Significance:** 3
**Empirical Novelty And Significance:** 2
**Recommendation:** 3
**Confidence:** 3

**Main Review:**

This paper studies adversarial attacks and defenses against algorithms for combinatorial optimization problems. The authors frame the problem, introduce RL-based algorithms for attack and defense, and present computational experiments.


**Summary Of The Paper:**

I will give the authors credit for introducing a novel problem to study, but they do not do a sufficient job motivating why the problem is an interesting or important one. At a high level I can buy the story they are trying to tell, but this really needs to be tied closely with a compelling (and, ideally, well studied) application  area. Without this, the claim in the introduction that "it is imperative to develop defense mechanisms" are unjustified. It is not clear to me that scheduling or TSP are problems where you would typically encounter adversarial inputs, and if the authors have concrete cases to illustrate this, please highlight them. If Fraud Coverage is a well-studied or established problem, provide citations to this effect.

Beyond the motivation of the problem itself, it is also not clear to me why the attack model studied in the paper is the "right" one. The authors should spend more time making a convincing case why their model is a plausible one. Again, this justification will need to be tied with a (compelling) application.

Other comments:
* p2: "Combnaotorial"
* The formalism for combinatorial optimization presented on page 3 is unlike any I have seen before. How should we understand what the x variables correspond to: binary indicators on e.g. edges? If so, where are the explicit binary constraints? It is also not clear to me what it means for the constraints to be "usually encoded in graphs".
* What does it mean to "loosen part of the constraint h_i"? Please introduce more formality about what h_i are, what loosen means, etc.
* I find it interesting that the FC attack against Gurobi essentially seems to find ways to make the problems extremely difficult to solve, as opposed to degrading the solution quality (though that may also occur). I wonder if the authors can gain any insight into what the attack is doing here.

**Summary Of The Review:**

The authors did not do a sufficient job motivating why the problem or model were interesting or important.

---

> ### Author Response · Authors · 2021-11-10
> **Response to Review.**
>
> Thank you for your time and comments. Our responses are as follow:
> - For the motivation, above all, the paper's main target falls on the CO solvers' robustness, but not deliberate attack and defensive methods. As we have mentioned in the introduction, CO problems are usually sensitive to perturbation due to their discrete and combinatorial nature. So for practical problems like DAG and Fraud Coverage, whose constraints (i.e. inherent graph structure) will evolve over time, they may indeed encounter adversarial scenarios at some point (our experiments have shown that even random perturbation to the input could poison the solvers to some extent). So to find out these adversarial inputs and develop corresponding defensive methods, we propose the general ROCO framework in the paper.
> - For the attack model, let me tell its plausibility along with the meaning of "loosen part of the constraints $h_i$". Here $h_i$ stands for a particular constraint in the CO problem. For example, in the DAG problem, $h_i$ could be a directed edge that ensures a certain task could only be executed after another one. So if we loosen part of the constraints, the problem's feasible solution space is expanded, yielding a better (at least the same) theoretical optimal solution than the original one. Specifically, in our setting, a successful attack is defined as: Loosen part of the constraints of the original CO problem, re-run the solver on the new problem but get a worse solution. The attack could reveal the vulnerability of the solver for that the solver could not adapt to the new scenario (with a better optimal solution) and give an unreasonable worse solution.
> - For formalism for combinatorial optimization (i.e. Eq(1)), we use a somewhat abstract and not that common-used formulation. Here $x$ denotes the general decision variable, and its specific form could vary with different CO problems. For example, in TSP, $x$ is a permutation of the cities to travel, while in Fraud Coverage, $x$ is the set of rules we choose to cover the fraudulent transactions. As mentioned before, the set $\{h_i\}$ stands for constraints, and they are usually encoded in graphs like DAG's constraints could be represented by a directed graph.
> - Finally, Fraud Coverage is a novel CO problem (we also prove its NP-hardness in the appendix) we discover in real life along with real-world dataset, so please forgive us for not being able to provide more citations. And for the FC attack against Gurobi, we set time limit in our code and find our attack model extremely effective (the solver could not even give a feasible solution after attacked) under certain settings. We also do experiments in Gurobi's solving time and the results are shown in Figure 4. Besides, we are looking forward to specifically attacking Gurobi's solving time as our future work. However, the theoretical analysis could be hard since Gurobi is a commercial solver without open source code.
> **You could also refer to our official comments for more details.** We sincerely hope that our explanations can answer your questions and help you understand our work better. We are looking forward to further discussions to discover more pros and cons of this paper.

---

### Author Response · Authors · 2021-11-10
**Clarify some misunderstandings about this article.**

- First, let me re-introduce our motivation. This paper mainly focuses on CO solvers' robustness, but not deliberate attack and defensive methods. It's common sense that CO problems are usually sensitive to perturbation due to their discrete nature. So for practical problems like DAG and Fraud Coverage, whose constraints will evolve over time, they may indeed encounter adversarial scenarios at some point (our experiments have shown that even random perturbation could poison the solvers to some extent). To find out these adversarial inputs and develop corresponding defensive methods, we propose the ROCO framework in this paper.
- Second, we re-state our attack and defensive models here. It's not realistic to get CO problems' ground truth optimal solution due to their NP-hardness. So we bypass this and come up with another comparison method to reveal the solvers' vulnerability.
  - For attack, suppose we loosen part of the constraints of a CO problem, then the feasible solution space will be expanded and we will get a better (at least the same) theoretical optimal solution to the new problem (and we will expect our solver to get a better solution to the new problem). However, some CO solvers may not adapt to this kind of revision and will get an even worse solution than the original one, and that is what we called non-robustness.
  - For defense, we adopt the similar idea to the attack. We add additional constraints to the original problem, thus the new solution (under stronger constraints) is absolutely feasible for the original one. In this way, we keep modifying the graph structure (i.e. constraints) and running the solver again to get a new solution, adapting it to the original problem to get a better solution value (although the optimal solution could not be better than the original one, the solver solution could get better under an easy-to-solve graph structure).
- Finally, it's our negligence that we haven't explained our formulation well in the paper. Here we do some compensated explanations to understand our settings better. In Eq(1), we first introduce $f(x|G)$, where $x$ is a solution to the problem and $f$ is the value function (mapping a solution to a value) of a certain CO problem. Besides, as we've described in the paper, $G$ is the graph that encodes the problem constraints (e.g. The DAG problem could be encoded by a directed graph), and $f(x|G)$ means we calculate the solution value of $x$ on graph $G$. We emphasize "on graph $G$" here because we may use the solution on different graphs. For example, we use $f(S(Q')|G')$ in attack to compare the solver's performance in different problem settings, but use $f(S(Q')|G)$ during defense to get a better solution value in the original problem setting.

---

### Decision · Program_Chairs · 2022-01-20

**Decision:**

Reject

**Comment:**

The paper aims at developing mechanisms for adversarial attack and defense towards combinatorial optimization solvers, where the solver is treated as a black-box function and the original problem’s underlying graph structure is attacked under a given budget. While the reviewers found the problem novel and interesting, they are not convinced by the problem formulation and the proposed solutions, as well as the experimental setup. Some of the points that the reviewers brought up during the discussion include: (i) the attack to the TSP does not follow the main paper's attack principle of adding and deleting edges, (ii), in general, it has not been explained why all these modification are really "relaxations", (iii) the notations are very confusing, and (iv) while authors' response on loosening the constraints makes sense, but the experiments (i.e., the TSP problem setting) in this work are not consistent with such clarification. Addressing the above points will significantly improve the manuscript.